

# Nonlinear Luttinger liquid: exact result for the Green function in terms of the fourth Painlevé transcendent

Tom Price[1,2*], Dmitry L. Kovrizhin[1,3,4] and Austen Lamacraft[1]

1 TCM Group, Cavendish Laboratory, University of Cambridge, J.J. Thomson Ave.,
Cambridge CB3 0HE, UK
2 Institute for Theoretical Physics, Centre for Extreme Matter and Emergent Phenomena,
Utrecht University, Princetonplein 5, 3584 CC Utrecht, The Netherlands
3 National Research Centre Kurchatov Institute, 1 Kurchatov Square, Moscow 123182, Russia
4 The Rudolf Peierls Centre for Theoretical Physics, Oxford University, Oxford, OX1 3NP, UK

* t.a.price@uu.nl

## Abstract

We show that exact time dependent single particle Green function in the Imambekov–Glazman theory of nonlinear Luttinger liquids can be written, for any value of the Luttinger parameter, in terms of a particular solution of the Painlevé IV equation. Our expression for the Green function has a form analogous to the celebrated Tracy–Widom result connecting the Airy kernel with Painlevé II. The asymptotic power law of the exact solution as a function of a single scaling variable $x/\sqrt{t}$ agrees with the mobile impurity results. The full shape of the Green function in the thermodynamic limit is recovered with arbitrary precision via a simple numerical integration of a nonlinear ODE.

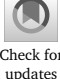

## 1 Introduction

The theory of Luttinger liquids has been extremely successful in providing an effective microscopic description of low energy equilibrium properties of one dimensional quantum systems [1–3]. Its approximation of the free fermion dispersion with a linear one leads to a quadratic theory in terms of bosonic quasiparticles, making the calculation of correlation functions a textbook exercise. However, this same approximation produces a catastrophic failure of the theory when one is concerned with time dependent properties, the simplest example of which is the single particle Green function (GF) [4,5]. This situation is clearest in chiral systems, such as quantum Hall edge states, where the linearization of dispersion results in a spacetime dependence of two–point correlators solely on $x - v_F t$, and concomitant $\delta$–function behaviour of the spectral function.

To cure this sickness, a phenomenological "nonlinear Luttinger liquid" theory has been developed, beginning with Ref. [6] and reviewed in Ref. [4]. A "mobile impurity" couples to the Luttinger liquid and resolves the degeneracy of the Luttinger spectrum to capture the correct analytical structure of dynamical correlation functions. Combined with exact solutions the mobile impurity model can give *exact* power law exponents of correlation functions at threshold [7,8].

At low energies Ref. [5] identified a scaling regime where the spectral function is determined by a one–parameter family of functions $D_\eta(s)$, where $\eta$ is fixed by the Luttinger parameter $K$, and the scaling variable $s = (\omega - v_F k)/(k^2/2m)$. Ref. [5] were able to find the exact power law behaviour for $D_\eta(s)$ at the thresholds $s = \pm 1$. We stress that not only is the threshold behaviour universal, in the sense of being determined solely through the Luttinger parameter and not on any other microscopic details, but so is the entire functional form of $D_\eta(s)$. However, away from the thresholds, the exact shape of the universal function $D_\eta(s)$ cannot be calculated within the mobile impurity model, and one has to resort to evaluation of fermionic determinants. This was carried out in the pioneering Imambekov–Glazman paper [5], and requires working in a truncated Hilbert space. More importantly because they are based on numerical calculations they naturally fail to reveal the analytic structure of the universal function, which as we show below is described by a fourth Painlevé transcendent. The latter equation is one of the celebrated nonlinear equations of mathematical physics and has applications in many different contexts including random matrix theory, symmetry reductions of integrable partial differential equations, and quantum gravity [9].

In this work we show that the Green function $G(x, t)$ can be written exactly in the thermodynamic limit for any $(x, t)$ in terms of a scaling function $g(\sigma)$ with $\sigma = x\sqrt{m/t}$. Here, the function $g(\sigma)$ is related to the fourth Painlevé transcendent, and $x$ is the comoving coordinate with the speed of sound to account for any linear dispersion. We solve the nonlinear ODE numerically specifying Luttinger liquid initial conditions at $t = 0$, which gives the Green function shown in Fig. 1.

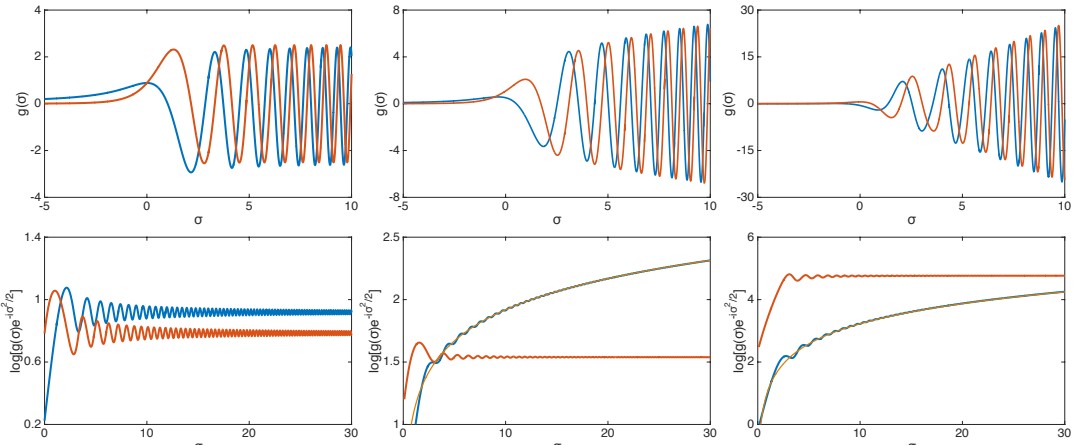

Figure 1: Top row: real (blue) and imaginary (red) parts of the scaling function $g(\sigma) = (t/m)^{\eta^2/2} G(x,t)$, obtained by solving numerically the ODE, see Appendix C; (left column) free fermions $\eta = 1$, (central column) $\eta = 1.2$, (right column) $\eta = \sqrt{3}$ which corresponds to $\nu = 1/3$ FQHE case. Bottom row: the logarithm of the scaling function $\log[g(\sigma)e^{-i\sigma^2/2}]$ for the same values of $\eta$, compared for $\eta = 1.2$, $\eta = \sqrt{3}$ with the asymptotic power law dependence Eq. (28) (orange line). The power law follows from the mobile impurity asymptotics [5] we give in Eqn. (28) for $G(x,t)$, which imply for $\sigma = \sqrt{m v^2 t} > 0$, $g(\sigma) \sim \sigma^{\eta^2 - 1 - 2(\eta-1)^2} e^{i\sigma^2/2}$, and for $\sigma < 0$, $g(\sigma)$ is asymptotic to the Luttinger power.

## 2 Fredholm determinant

*Definitions.* There is strong evidence [5,10,11] that low energy eigenstates of generic chiral 1D models with short range interactions can be put in correspondence with those of free fermions with quadratic dispersion generated by the Hamiltonian

$$\hat{H} = \sum_k \frac{1}{2m}(k + k_F)^2 : \hat{c}_k^\dagger \hat{c}_k :, \tag{1}$$

The linear dispersion $(k_F/m)k$ is the chiral Luttinger Hamiltonian. Quadratic dispersion is the leading irrelevant operator, and the unique local chiral operator at this scaling dimension, which is the essence of the argument identifying the universal scaling regime [5]. Of course, this does not mean that the correlators of physical operators are trivial. The bosonization formula shows free fermions $c^\dagger(x) \sim e^{i\varphi^\dagger(x)}$ of Eqn. (1) are not equal to the physical fermions $e^{i\eta\varphi^\dagger}$. The chiral boson $\varphi$ is defined below, Eqn. (2), and $\eta$ is related to Luttinger parameter by $\eta = (K^{1/2} + K^{-1/2})/2$, or for the FQHE edge states at filling fraction $\nu$ is given by $\eta = 1/\sqrt{\nu}$. It accounts for interactions between physical fermions [1–3]. It is also worth noting that there are known cases when a quadratic fermion dispersion is forbidden by a symmetry, e.g. a spin-$\frac{1}{2}$ chain in zero field [12], where the dispersion starts with a cubic term. Here we focus on the case of quadratic dispersion. Defining bosonic fields

$$\hat{\varphi}^\dagger(x) = -\sum_{q>0} \frac{1}{\sqrt{n_q}} \hat{b}_q^\dagger e^{-iqx - qa/2}, \; q = \frac{2\pi}{L} n_q, \; n_q \in \mathbb{Z} \tag{2}$$

where $L$ is the length of the system, $a$ is the large momentum cutoff, and $\hat{b}_q^\dagger$ is the boson creation operator given in terms of fermions as

$$\hat{b}_q^\dagger = \frac{i}{\sqrt{n_q}} \sum_{k=-\infty}^{\infty} \hat{c}_{k+q}^\dagger \hat{c}_k, \tag{3}$$

one can write the time dependent single particle GF as

$$G(x,t) = (2\pi/L)^{\eta^2} \langle e^{-i\eta\varphi(x,t)} e^{i\eta\varphi^\dagger(0,0)} \rangle. \tag{4}$$

Here the average is taken with respect to a bosonic vacuum. The only difference from the standard bosonization expression is that instead of taking a linear fermionic spectrum, time evolution is generated by Eqn. (1). Using Lehmann representation in terms of particle–hole states the Green function can be written [13,14]

$$G(x,t) = (2\pi/L)^{\eta^2} \sum_{n=0}^{\infty} \sum_{p_i,q_j} \det\left[ L(p_i,q_j|x,t) \right]_{i,j=1}^{n}. \tag{5}$$

Here, the kernel $L(p,q|x,t)$, defined in Eqs. (6), (7), is the matrix element of a vertex operator between the filled Fermi sea, and a single particle–hole pair with quantized momenta $p_i, q_j$ correspondingly, derived in Ref. [14]. From Eq. (5) it is possible to find exact power law singularities in the spectral function [15,16]. However, an explicit evaluation of the determinant has not been addressed so far.

In the thermodynamic limit the kernel assumes the "integrable" form [17,18]

$$L(p,q|x,t) = \frac{f(p|x,t) \cdot g(q|x,t)}{p-q}, \tag{6}$$

where the dot product acts on the $r,s$ indices in $f_r(k)$, $g_s(k)$,

$$f_1(k) = g_2(k) = \theta(k) \frac{(kx)^\eta}{\Gamma(\eta)} e^{-\frac{i}{2}\left[\frac{k^2}{2m}t - k(x+ia)\right]},$$

$$f_2(k) = g_1(k) = \theta(-k) \frac{(|k|x)^{-\eta}}{\Gamma(1-\eta)} e^{\frac{i}{2}\left[\frac{k^2}{2m}t - k(x+ia)\right]}. \tag{7}$$

The vanishing of $L(k_1,k_2)$ when $k_1,k_2$ are of the same sign ensures that only states with equal numbers of particles and holes contribute. The kernel Eq. (6) at $t=0$ and $x$ imaginary describes a probability distribution on the Young diagrams, which are in a one to one correspondence with many body states $\lambda$, known as the mixed $z$–measure [19].

By definition the right hand side of the Green function, Eqn. (5), is a Fredholm determinant $G(x,t) = (2\pi/L)^{\eta^2} \det[1 + L(x,t)]$, which we now study with Riemann–Hilbert methods [18]. We note that in some cases a simple direct numerical evaluation of Fredholm determinants is possible by using an $m$ point quadrature rule to estimate the integrals appearing in the Fredholm expansion, which reduces the problem to the evaluation of an $m \times m$ determinant [20]. However, a quadrature rule cannot be directly applied to the highly oscillatory kernel (6), (7), which prohibits direct application of this method.

## 3 Riemann–Hilbert problem and differential equations

In this section we describe how to find a set of nonlinear ordinary differential equations from the Fredholm determinant representation of the Green function, Eqn. (5). First, we take a log derivative of GF with respect to $x$,

$$\partial_x \log G(x,t) = \mathrm{Tr}\,(1+L)^{-1} \partial_x L. \tag{8}$$

Now we exploit the integrable form of the kernel (6). First, direct evaluation of $\partial_x L(k_1,k_2|x,t)$ shows the right hand side is $\frac{i}{2} \mathrm{tr}(1+L)^{-1} f(k_1|x,t)\sigma_3 g(k_2|x,t)$. Importantly, the resolvent $K = L(1+L)^{-1}$ is also integrable [17,21]:

$$K(k_1,k_2) = \frac{F(k_1) \cdot G(k_2)}{k_1 - k_2}, \tag{9}$$

where $F = (1 + L)^{-1} f$, $G^T = g^T (1 + L)^{-1}$. Then we can express the log derivative of the Green function as a trace over both the momentum $k$ and the $2 \times 2$ matrix structure of $f_i, g_j$

$$\frac{\partial}{\partial x}\bigg|_t \log \det [1 + L(x, t)] = \frac{i}{2} \operatorname{Tr} F_i(k) \sigma_3^{ij} g_j(k). \tag{10}$$

The object on the r.h.s. can be found by solving a Riemann–Hilbert problem (RHP). It is first convenient to rescale momenta $\lambda = k \sqrt{t/m}$ so that $f, g$ are functions of $\lambda$ and $\sigma$. We will only discuss the retarded Green function so that $t \geq 0$. To build the RHP we define the $2 \times 2$ matrix–valued function $m(z)$ in terms of known $g$ and unknown $F$,

$$m(z) = 1 - \int_{-\infty}^{\infty} d\lambda \, \frac{F(\lambda) g^T(\lambda)}{z - \lambda}, \tag{11}$$

which is analytic in the complex plane except for the real axis. Plemelj formula and the relation $F = (1 + L)^{-1} f = mf$ shows that the limiting values above and below the real axis satisfy the jump condition

$$m^+(\lambda) = m^-(\lambda) \big[ 1 + 2\pi i f(\lambda) g^T(\lambda) \big]. \tag{12}$$

At large $z$, $m \sim 1 + m^{(1)}/z + \dots$ approaches the identity as

$$m(z) \sim 1 - z^{-1} \int_{-\infty}^{\infty} d\lambda \, F(\lambda) g^T(\lambda) + O(z^{-2}). \tag{13}$$

The jump equation Eqn. (12) and behaviour $m(z) \to 1$ at infinity fix $m(z)$ uniquely in terms of the known functions $f, g$. What's more, if we can solve this RHP for $m(z)$, comparing Eqns. (13), (10), we need only extract its residue at infinity to express the log derivative of $G(x, t)$ as a $2 \times 2$ trace

$$\frac{\partial}{\partial \sigma}\bigg|_t \log \det [1 + L(x, t)] = -\frac{i}{2} \operatorname{tr}[m^{(1)} \sigma_3]. \tag{14}$$

Explicit solutions to matrix RHPs are rare. For the equal time case, the solution is known explicitly in terms of Whittaker functions [22] from studies of $z$–measures on Young diagrams, and is presented here in Appendix B. However, by Eqn. (14), the task of finding $G(x, t)$ does not require a full solution of the RHP, but only its residue $m^{(1)}$. The strategy is to derive a Lax equation, which lead to consistency conditions for matrix elements of $m^{(1)}$. The consistency conditions are given by a system of two second order coupled nonlinear equations. Then we use the first integral to show that this system is equivalent to Painlevé IV. The initial conditions for this equation are obtained from asymptotic data as $\sigma \to +i\infty$, which corresponds to GF at $t = 0$.

The connection we find between the Fredholm determinant and Painlevé IV is analogous to the Tracy–Widom result in random matrix theory. The Fredholm determinant of the Hermite kernel, restricted to a finite interval $(0, s)$, describes the distribution function of the largest eigenvalue in the $N \times N$ Gaussian Unitary Ensemble. Its second log derivative satisfies Painlevé IV and in the edge scaling limit reduces to Painlevé II [21]. The main difference between the two comes from initial conditions. This is rather fortunate because in our case the numerical calculation does not require considerable efforts, as opposed to the Tracy–Widom problem where the required Hastings–McLeod solution is inherently unstable [23, 24]. Below we outline the connection of the Fredholm determinant with the fourth Painlevé transcendent. For details see Appendix A.

### 3.1 Lax equation

The matrix $\Psi(\lambda, \sigma) \equiv m(\lambda, \sigma)e^{-\Theta\sigma_3/2}$, where $\Theta = i\lambda^2/2 - i\sigma\lambda - 2\eta\log(\lambda\sigma)$, satisfies a RHP with a piecewise constant jump matrix. By differentiating the jump equation for $\Psi$ we find that $\partial_\lambda\Psi \cdot \Psi^{-1}$ has a simple pole in $\lambda$ at the origin, and $\partial_\sigma\Psi \cdot \Psi^{-1}$ is an entire function of $\lambda$. On the other hand we can expand $m = 1 + m^{(1)}(\sigma)/\lambda + \dots$ to calculate directly $\partial\Psi \cdot \Psi^{-1}$. Combined with our knowledge of the analyticity of $\Psi$ we can terminate the expansions to find expressions for $\partial_\sigma\Psi \cdot \Psi^{-1}$ and $\partial_\lambda\Psi \cdot \Psi^{-1}$ explicitly in terms of $m^{(1)}(\sigma)$ and $\lambda$. The matrix $m^{(1)}$ contains the derivative of the Green function on its diagonal by Eqn. (14). Since $\det m = 1$, $m^{(1)}(\sigma)$ is traceless and we parameterize it

$$m^{(1)}(\sigma) = \begin{pmatrix} i\partial_\sigma \log G(x,t) & \sigma^{2\eta}e^{i\sigma^2/2}\bar\zeta(\sigma) \\ \sigma^{-2\eta}e^{-i\sigma^2/2}\zeta(\sigma) & -i\partial_\sigma \log G(x,t) \end{pmatrix}. \tag{15}$$

The absence of a pole at $\lambda = 0$ in $\partial_\sigma\Psi \cdot \Psi^{-1}$ implies the key relation

$$\frac{\partial^2}{\partial\sigma^2} \log G(x,t) = -\bar\zeta\zeta. \tag{16}$$

It is then straightforward but tedious to cross differentiate $\partial_\sigma\Psi \cdot \Psi^{-1}$, $\partial_\lambda\Psi \cdot \Psi^{-1}$ and show that $\zeta, \bar\zeta$ satisfy nonlinear Schrödinger type equations

$$\begin{aligned} 0 &= -\bar\zeta'' - i\sigma\bar\zeta' + 2i\eta\bar\zeta + 2\bar\zeta\zeta\bar\zeta, \\ 0 &= -\zeta'' + i\sigma\zeta' + 2i\eta\zeta + 2\zeta\bar\zeta\zeta. \end{aligned} \tag{17}$$

The two equations are related by complex conjugation and sending $\eta \to -\eta$, and we emphasize that $\bar\zeta \neq \zeta^*$. Next, we shall see how to find a first integral of this pair of equations and show that $\zeta^{-1}\partial\zeta$ and $\bar\zeta^{-1}\partial\bar\zeta$ satisfy Painlevé IV.

*First integral.* We have the first integral of the NLSEs

$$\zeta'\bar\zeta' + \left(\eta - i\bar\zeta\zeta\right)^2 = 0. \tag{18}$$

Differentiating the left hand side and using Eqs. (17) shows that this quantity is indeed invariant under the time evolution, and comparison with the equal time result, see Eqn. (35) of Appendix B and Ref. [22], fixes it to be zero at all times.

### 3.2 Painlevé IV

The system Eqs. (17) and their first integral, Eqn. (18), reduces to PIV for $\varphi \equiv \zeta^{-1}\partial_\sigma\zeta$. Substitute $\bar\zeta\zeta = \frac{1}{2}\varphi' + \frac{1}{2}\varphi^2 - \frac{i}{2}\sigma\varphi - i\eta$ from Eqs. (17) into the Eq. (18). The result is a special case of the PIV [25] equation for $\varphi \equiv \zeta^{-1}\zeta_\sigma$

$$\frac{\varphi''}{\varphi} = \frac{1}{2}\left(\frac{\varphi'}{\varphi}\right)^2 + \frac{3}{2}\varphi^2 - 2i\sigma\varphi - \frac{1}{2}\sigma^2 - i[2\eta - 1]. \tag{19}$$

Similarly $\bar\zeta^{-1}\partial\bar\zeta$ satisfies the conjugated equation with $\eta \to -\eta$. Painlevé IV can be brought to the more intuitive form of a nonlinear harmonic oscillator when we put $\rho = e^{-i\pi/4}\sigma$, $\zeta^{-1}\frac{d\zeta}{d\rho} = 2u^2(\rho)$, so that

$$\frac{d^2u}{d\rho^2} = 3u^5 + 2\rho u^3 + \left(\frac{1}{4}\rho^2 + \eta - \frac{1}{2}\right)u. \tag{20}$$

We can also find a differential equation directly for $c = i\partial_\sigma \log G$. To do so, we need to use another relation that we obtain from the vanishing of the $1/z^2$ terms in the expansion of $\partial_\sigma \Psi \cdot \Psi^{-1}$ and $\partial_z \Psi \cdot \Psi^{-1}$, which implies

$$0 = c - \sigma c' - \bar{\zeta}\zeta' + \bar{\zeta}'\zeta. \tag{21}$$

Combining this with Eqn. (16), NLSEs (17), and their first integral (18), it can be shown that $c$ satisfies the equation

$$0 = (c'')^2 + (\sigma c' - c)^2 - 4ic'(c' + \eta)^2. \tag{22}$$

This equation is known as the Jimbo–Miwa $\sigma$ form of Painlevé IV (the terminology is an unfortunate clash of notation), see Ref. [26]. In their language, our result is therefore that the nonlinear Luttinger liquid Green function *is* the Miwa–Jimbo tau function for Painlevé IV.

## 3.3 Boundary conditions and final result

To find the Green function $G(x, t)$ we must twice integrate the "amplitude" $\bar{\zeta}\zeta$ of the equations (17). We fix the boundary conditions for the NLSEs by demanding that $G(x, t) \to G(x, 0)$ at short times. Since we work in the similarity variable $\sigma = x\sqrt{m/t}$ we have to identify the region in the complex plane where large $\sigma$ corresponds to the short time limit. To warm up, consider first the free fermion problem, where the Green function

$$\frac{G(x, t)}{G(x, 0)} = -i\sigma \int_0^\infty d\lambda \, e^{-i[\lambda^2/2 - \lambda\sigma]}. \tag{23}$$

When $\sigma$ is large and approaches positive real axis from above, the integral is dominated by the saddle point at $\lambda = \sigma$

$$\frac{G(x, t)}{G(x, 0)} \sim -\sqrt{2\pi}e^{i\pi/4}\sigma e^{i\sigma^2/2}, \tag{24}$$

where the oscillations reflect that on the supersonic side the correlation function is dominated by particle excitations. For $\sigma = i\chi$ large and imaginary, the integrand has already exponentially decayed by the time the oscillations kick in, and the integral is dominated by its behaviour at $\lambda \ll 1$. We may approximate it by dropping the quadratic term and

$$\frac{G(x, t)}{G(x, 0)} \sim \chi \int_0^\infty d\lambda \, e^{-\lambda\chi} = 1, \tag{25}$$

recovering the equal time behaviour. Analogously when $\eta \neq 1$ and $\sigma = i\chi$ is large and imaginary oscillations in the Fredholm determinant may be neglected, and the Green function approaches the equal time correlator. Using the asymptotic result for $\sigma_0$ large and imaginary $\log G(\sigma_0) \sim -\eta^2 \log \sigma_0$ and taking the limit $\sigma_0 \to i\infty$ we arrive at the main result of this paper

$$\frac{G(x, t)}{G(x, 0)} = \exp\left(-\int_{i\infty}^\sigma d\sigma' \left[(\sigma - \sigma')\bar{\zeta}\zeta(\sigma') - \frac{\eta^2}{\sigma'}\right] + \eta^2\right). \tag{26}$$

Here $\bar{\zeta}$, $\zeta$ solve the NLSEs (17), and their log derivatives satisfy Painlevé IV. The asymptotic behaviour for large imaginary $\sigma$ is given by the linear Luttinger problem, where the off diagonal elements of $m^{(1)}$ in Eqn. (15) behave as powers, see Appendix B. Owing to Gaussian factors in our parameterization of $m^{(1)}$, the asymptotic behaviour for the Painlevé IV solutions is $\varphi = \zeta^{-1}\partial\zeta \sim i\sigma$, $\bar{\varphi} = \bar{\zeta}^{-1}\partial\bar{\zeta} \sim -i\sigma$. As far as we are aware this solution has not been studied (see Refs [27, 28] for recent reviews of Painlevé IV). The solutions obtained by integration of an equivalent set of first order equations C, for the free fermion $\eta = 1$, $\eta = 1.2$, and $\eta = \sqrt{3}$ corresponding to the $\nu = 1/3$ Laughlin state, are plotted in Fig. 1.

### 3.4 Mobile Impurity Asymptotics

For the Green function considered here the singularities in $\omega, k$ space from Ref. [5] are *exact*. The retarded Green function studied here is related to the Imambekov–Glazman $D_\eta$ function by Fourier transform

$$G(x,t) = \theta(t) \int_0^\infty dk \, k^{\eta^2-1} e^{ikx} \int_{-\infty}^\infty ds \, e^{-i\frac{k^2 t}{2m}s} D_\eta(s).$$  (27)

At the thresholds of support $s = \pm 1$, $D_\eta(s)$ has singularities $(1 \mp s)^{[\eta \mp 1]^2 - 1}$ [5]. The shift of the power $\eta \to \eta - 1$ from the Luttinger liquid exponent follows from the exact degeneracy when $L \to \infty$ among all states with a single "hard" particle that carries all the energy, and soft modes carrying no energy but a finite momentum. Because one has to use $e^{-i\varphi}$ of the vertex operator $e^{-i\eta\varphi}$ to create the hard particle, the exponent of the Luttinger liquid is shifted by one. This can also be seen in the form factors [15, 16]. These singularities in $D_\eta(s)$ yield asymptotics for the Green function at large $x, t$ keeping the ratio $v = x/t$ fixed,

$$G(x,t) \sim \begin{cases} \left(i\frac{1}{2}mv^2 t\right)^{-[\eta-1]^2-1/2} e^{i\frac{1}{2}mv^2 t}, & x > 0, \\ (-i[x+i0])^{-\eta^2}, & x < 0. \end{cases}$$  (28)

In Fig. 1 we compare the supersonic asymptotic with a numerical evaluation of the Green function obtained by solving the differential equations.

## 4 Conclusion

To conclude, we have shown that the Green function in the chiral nonlinear Luttinger liquid may be expressed in terms of the fourth Painlevé transcendent in the similarity variable $x\sqrt{m/t}$, dependent only upon the Luttinger parameter. The solution may be found numerically, and the real space Green function shows a power law and oscillations characteristic of the edge singularity in the spectral function. The Green function here for $z = x + ib$ complex describes propagation of a Leviton $|z\rangle = e^{-i\eta\varphi(z)}|0\rangle$ created by an application of a Lorentzian voltage pulse of width $b/v_F$ along a 1D wire [29], as has been studied in recent experiments [30]. It is typically assumed in the literature that propagation of such states is *ballistic* – see however Ref. [13]. An interesting extension of this work, which is relevant to recent experiments [30, 31], would be to identify clear signatures of the dispersion in the interference of two Leviton pulses. It is also interesting to wonder whether other nonequilbrium correlation functions may be tackled by this method. Nonlinear partial differential equations are known [32, 33] for the correlation function $\text{Tr}\left[e^{-i\eta\varphi(x,t)}e^{i\eta\varphi^\dagger(x',t')}\rho\right]$, where $\rho$ is the exponential of fermion bilinears but otherwise arbitrary. When $\rho$ lacks any length scale, as in the case here where it simply projects onto the ground state, the PDEs must reduce to Painlevé IV.

## Acknowledgements

We acknowledge discussions with F. Essler.

**Funding information** We acknowledge support from EPSRC. D.K. is supported by EPSRC Grant No. EP/M007928/1.

## A  Lax pair

The jump matrix $v$ for $m$ is $\lambda$ and $\sigma$ dependent via the function $\Theta = i\frac{\lambda^2}{2} - i\sigma\lambda - 2\eta\log\lambda\sigma$, where the logarithm has a branch cut on the negative real axis, and

$$v = \begin{cases} 1 + \frac{2\pi i}{\Gamma(\eta)^2}e^{-\Theta(\lambda)}\sigma_+, & \lambda > 0, \\ 1 + \frac{2\pi i}{\Gamma(1-\eta)^2}e^{\Theta^-(\lambda)}\sigma_-, & \lambda < 0. \end{cases} \tag{29}$$

The matrix $\Psi = me^{-\Theta\sigma_3/2}$ satisfies a Riemann–Hilbert problem with a piecewise constant jump matrix $\tilde{v}$

$$\tilde{v} = \begin{cases} 1 + \frac{2\pi i}{\Gamma(\eta)^2}\sigma_+, & \lambda > 0, \\ e^{2\pi i\eta\sigma_3} + e^{2\pi i\eta}\frac{2\pi i}{\Gamma(1-\eta)^2}\sigma_-, & \lambda < 0. \end{cases} \tag{30}$$

Taking determinants of the jump equation we find $(\det m)^+ = (\det m)^-$, and combined with the limiting value $\det m \to 1$ at infinity, Liouville's theorem says that $\det m = 1$ everywhere. Consequently $m^{-1}$ and $\Psi^{-1}$ are also analytic away from the real axis. From the definition $\Psi = me^{-\Theta\sigma_3/2}$ we find the $\sigma$ derivative

$$\partial_\sigma\Psi \cdot \Psi^{-1} = m_\sigma m^{-1} + \left(\frac{iz}{2} + \frac{\eta}{\sigma}\right)m\sigma_3 m^{-1}. \tag{31}$$

Expanding $m = 1 + z^{-1}m^{(1)} + z^{-2}m^{(2)} + \dots$ and organizing terms order by order in $z^{-1}$, we find $\partial_\sigma\Psi \cdot \Psi^{-1}$ is

$$\frac{i}{2}z\sigma_3 + \begin{pmatrix} \eta/\sigma & -i\sigma^{2\eta}e^{i\sigma^2/2}\bar{\zeta} \\ i\sigma^{-2\eta}e^{-i\sigma^2/2}\zeta & -\eta/\sigma \end{pmatrix} + O(z^{-1}). \tag{32}$$

We have used the parameterization Eqn. (15) for $m^{(1)}$. However, independence of the jump matrix for $\Psi$, Eqns. (30), on $\sigma$ tells us that $[\partial_\sigma\Psi \cdot \Psi^{-1}]^+ = [\partial_\sigma\Psi \cdot \Psi^{-1}]^-$ and is therefore an entire function, so all terms in the big $O$ must vanish. Vanishing of the $z^{-1}$ term tells us that $dc/d\sigma = -i\bar{\zeta}\zeta$, which will now use in calculating $\partial_z\Psi \cdot \Psi^{-1}$. Proceeding as before we find $\partial_z\Psi \cdot \Psi^{-1}$ is

$$-\frac{i}{2}z\sigma_3 + \begin{pmatrix} i\sigma/2 & i\sigma^{2\eta}e^{i\sigma^2/2}\bar{\zeta} \\ -i\sigma^{-2\eta}e^{-i\sigma^2/2}\zeta & -i\sigma/2 \end{pmatrix} + \frac{1}{z}\begin{pmatrix} \eta - i\bar{\zeta}\zeta & \sigma^{2\eta}e^{i\sigma^2/2}\bar{\zeta}_\sigma \\ \sigma^{-2\eta}e^{-i\sigma^2/2}\zeta_\sigma & -\eta + i\bar{\zeta}\zeta \end{pmatrix}. \tag{33}$$

Now there is a pole in $\partial_z\Psi \cdot \Psi^{-1}$ because the jump matrix for $\Psi$ is piecewise constant in $z$. This completes the derivation of the Lax pair, from which the nonlinear Schrödinger equations (17) follow.

## B  Exact solution to RHP for the Luttinger liquid

For completeness we give the exact solution to the RHP for the equal time problem in imaginary space $kx = i\chi$, first found in Ref. [22]. The RHP solution in the upper and lower half planes $m^\pm(\chi)$ is expressed in terms of Whittaker functions [25]

$$m^\pm(\chi) = \begin{pmatrix} e^{-\chi/2}\chi^{-\eta-\frac{1}{2}}W_{\eta+\frac{1}{2},0}(\chi) & \eta^2 e^{\chi/2}(e^{\mp i\pi}\chi)^{\eta-\frac{1}{2}}W_{-\eta-\frac{1}{2},0}(e^{\mp i\pi}\chi) \\ -e^{-\chi/2}\chi^{-\eta-\frac{1}{2}}W_{\eta-\frac{1}{2},0}(\chi) & e^{\chi/2}(e^{\mp i\pi}\chi)^{\eta-\frac{1}{2}}W_{-\eta+\frac{1}{2},0}(e^{\mp i\pi}\chi) \end{pmatrix}. \tag{34}$$

Using the analytic continuation formula for the Whittaker functions (Ref. [25], Eqn. 13.14.13) one can check that the jump equations (12), are satisfied for $t = 0$, and that $m(\chi) \to 1$ as

$\chi \to \infty$. The residue at infinity is

$$m^{(1)} = \begin{pmatrix} -\eta^2 & -\eta^2 \\ -1 & \eta^2 \end{pmatrix}. \tag{35}$$

Rescaling back to $x$, Eqn. (14) gives the Luttinger correlator in the form $\partial_x \log G = -\eta^2/x$. By Eqn. (33) the determinant of the $1/z$ coefficient of $\Psi_z \Psi^{-1}$ is the first integral of the NLSEs, Eqn. (18), and direct calculation using the solution (34) shows it vanishes.

## C   Numerical solution of differential equations

Numerically it is much quicker to solve a set of coupled first order nonlinear equations rather than directly attacking Painlevé IV. We make a gauge transform in the NLSEs $\zeta = ae^\gamma$, $\bar\zeta = \bar{a}e^{-\gamma}$, and choose $\gamma$ such that $\zeta_\sigma = e^\gamma(\bar\zeta\zeta + i\eta)$, $\bar\zeta_\sigma = e^{-\gamma}(\bar\zeta\zeta + i\eta)$ in order that the first integral, Eqn. (18), is automatically satisfied. The NLSE becomes a relation between $\gamma, a, \bar{a}$, and requires $\gamma' = a - \bar{a} + i\sigma$. Using this relation to eliminate $\gamma$ in the the expressions for $\varphi, \bar\varphi$ we get the coupled first order equations

$$\begin{aligned} a' + i\sigma a + a^2 - i\eta &= 2\bar{a}a, \\ \bar{a}' - i\sigma \bar{a} + \bar{a}^2 - i\eta &= 2\bar{a}a, \end{aligned} \tag{36}$$

that we found more convenient to solve than the Painlevé equation as it appears in Eqn. (20). For $\sigma$ large and imaginary $a, \bar{a}$ are small and non-oscillatory and we can look for series solutions in $1/\tau$, which we use to obtain the initial data $a(\sigma_0)$, $\bar{a}(\sigma_0)$. With the help of Mathematica we can pull out the coefficients of the asymptotic series, the first few terms of which are

$$\begin{aligned} a(\sigma) &\sim -i\frac{\eta}{\sigma} + \eta\frac{1-3\eta}{\sigma^3} + i\eta\frac{3+\eta(-11+18\eta)}{\sigma^5}, \\ \bar{a}(\sigma) &\sim i\frac{\eta}{\sigma} + \eta\frac{1+3\eta}{\sigma^3} - i\eta\frac{3+\eta(11+18\eta)}{\sigma^5}. \end{aligned} \tag{37}$$

We checked that the Green function obtained by solution of Eqns. (36) agrees with the exact result for the free fermion, Eqn. (23), when $\eta = 1$. For $\eta = 1$, 1.2, and $\sqrt{3}$ we plot the numerical solution in Fig. 1. We take the integration contour in Eqn. (26) to run from values around $75i$, down to the origin, and then along the real axis to the required value of $\sigma$. The plot of $\log[e^{-i\sigma^2/2}g(\sigma)]$ in Fig. 1 verifies both the power law and oscillations.

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
