# Peer review of "Nonlinear Luttinger liquid: Exact result for the Green function in terms of the fourth Painlevé transcendent"

_SciPost Physics, doi:SciPost Phys. 2, 005 (2017)_

## Round 2 · Referee Report · Anonymous · 2017-1-17

Strengths
1) Thorough analysis
2) Mathematical treatment of a physically relevant problem
3) Good presentation
Weaknesses
1) The result has formal rather than physical significance
Report
I have reviewed the manuscript by Price Kovrizhin and Lamacraft on the Painleve transcendents appearing in the theory of nonlinear liquids. The manuscript gives results on the integrable structure underlying the Green's function in interacting fermion gases, based on the theory by Glazman and Imambekov. The paper first formulates the problem as a Fredholm determinant with an integrable kernel, which in turns leads to a Riemann-Hilbert problem and lastly integrable ODEs identified as of the Painleve type.
The question of the interplay between classical integrable systems and fermionic observables is an important one, and the authors present a result which will bear some importance in this field.
The presentation is clear and avoids unnecessary complication to present the results.
I recommend publishing the paper in the Journal.
Requested changes
None.
Volker Meden on 2017-01-08 [id 84]
I would like to better understand what the third sentence of the Introduction is supposed to mean. Is the statement that time-dependent or dynamical (frequency-dependent) observables (e.g. observables derived from the time-dependent equilibrium single-particle Green function) of microscopic (e.g. lattice) models are unrelated to those computed within the Tomonaga-Luttinger model (which you characterize as being doomed)? It is generally believed that e.g. the low-energy power-law suppression of the momentum integrated single-particle spectral function of the Tomonaga-Luttinger model (as a function of frequency) can be found in microscopic models. Furthermore, the fourth sentence does not hold true if the two-particle interaction of a chiral Tomonaga-Luttinger model is considered to be momentum-dependent. This is e.g. discussed in Phys. Rev. B 47, 16205 (1993) (http://prola.aps.org/abstract/PRB/v47/i24/p16205_1). What does "catastrophic failure" refer to? What is the "sickness"? A momentum dependent interaction which decays to zero at large momenta renders the Tomonaga-Luttinger model a well defined model which does not suffer from any ultraviolet divergences.
Author: Tom Price on 2017-01-11 [id 85]
(in reply to Volker Meden on 2017-01-08 [id 84])We mean that frequency _and momentum_ resolved single-particle response functions of microscopic models are not described correctly by the Luttinger model (we note that the _low-frequency momentum-integrated_ spectral function is unaffected by nonlinear Luttinger liquid physics). See for example Ref [15] of the manuscript, which finds threshold power law behaviour for Lieb-Liniger model in agreement with the nonlinear Luttinger power laws of Ref [5].
We agree that our fourth sentence assumes a linear bosonic dispersion, and that allowing for a momentum-dependent boson interaction does generate a finite linewidth. The question is whether this captures the way the degeneracy of the linear Luttinger liquid is lifted in "generic" microscopic models. Comparison with the Lieb--Liniger model, as in Ref [15], or most simply the quadratically dispersing free fermion, reveals that fermionic dispersion is essential for capturing the true lineshape. For this reason we call the theory ''sick'', and the same problem is referred to as ''catastrophic failure''. These are of course not our original observations, but those which lead to the development of the nonlinear Luttinger liquid theory reviewed in Ref. [4].

---

## Round 2 · Referee Report · Anonymous · 2017-1-30

Strengths
1.) Exact and rigorous result
2.) Rather clear presentation
Weaknesses
1.) Analysis could have been carried our further
Report
The authors analyse a Fredholm determinant representation of a correlation function of
two vertex operators derived previously in [13,14]. This correlation function is of
some physical interest as it equals the single-particle Green function of a non-linear
extension of the Luttinger liquid paradigm of one-dimensional critical models with
central charge $c = 1$.
The kernel (6) of the integral operator is of integrable form. This implies [17] that
there is a general and well known way to associate a Riemann-Hilbert problem and an
integrable classical evolution equation with it. The authors follow this way and identify
the evolution equation as related to the fourth Painleve transcendent. They connect the
single-particle Green function with a special solution of this equation specified by
its asymptotics (eq. (26) and below). The authors obtain this solution numerically, insert
it into the formula for the single-particle Green function and compare with an asymptotic
expression obtained by phenomenological means in [5]. The agreement of numerical and
asymptotic results, shown in figure 1, is rather impressive.
Although I find the paper in general quite convincing, I have a few questions and suggestions
that might help to improve it further.
Requested changes
1.) I am puzzled by the fact that the numerator on the right hand side of (6) for $p = q$
in non-zero. This would be the case in all examples of kernels related to correlation function
that I know, where typically one has something like $g_2 (k) = - f_1(k)$ rather than
$g_2 (k) = f_1(k)$. Maybe the authors can comment on this point.
2.) (Small point) I would rather write "Explicit solutions of matrix RHPs are rare" below (14),
since scalar RHPs, which might be more familiar to the average reader, are easy to solve.
3.) Matrix RHPs connected with integrable integral operators are usually the starting point of
asymptotic analysis within the non-linear steepest descent approach of Deift and Zhou. I wonder
if this is expected to reproduce the result for the asymptotics of $G(x,t)$ obtained in [5], and
I would appreciate if the authors could comment on that.
4.) It has been shown (F. Bornemann, On the numerical evaluation of Fredholm determinants,
Mathematics of Computation 79 (2010), 871) that Fredholm determinants may sometimes be very
efficiently calculated by a direct numerical approach. A comment on this would be very welcome.
5.) It seems to me that the publication date in reference [18] is wrong.
6.) How about an extension to finite temperatures? In the example of the impenetrable
Bose gas considered e.g. in [18] such an extension is possible.
Thanks to the referee for their observations and helpful suggestions. In response to their requested changes:
-
The numerator of the Fredholm kernel in fact vanishes for $p=q$, and in fact more generally whenever $p$ and $q$ have the same sign, on account of the step functions. This is discussed after Eqn 7.
-
We agree and thank the referee for the edit.
-
A steepest descent calculation of the asymptotics must be slightly different from the usual case where only the saddle point (states in vicinity of the hard particle) contributes, since one has to account for soft particle--hole excitations at the Fermi point. This is to be contrasted with the asymptotic solution for the nonlinear Schrodinger equation, where the entire contribution comes from the saddle point. Such a calculation would therefore be interesting to do from a technical point of view but since (leading) asymptotics are already known exactly (ref [5], [15]) it is of limited physical interest. We point out again that in this problem the mobile impurity treatment of [5] is nothing but an exact summation of the Lehmann series for the spectral function in the vicinity of the threshold, and whilst the Hamiltonian in ref [5] is introduced phenomenologically the exponents governing asymptotic behaviour of its correlation functions are found exactly.
-
We thank the referee for the suggestion. Bornemann's method uses an $m$ point quadrature rule to estimate the integrals appearing in the Fredholm determinant, and as such cannot be directly applied to the highly oscillatory integrals appearing here.
-
We agree and thank the referee for spotting the typo.
-
Extending the approach here to finite temperature is an interesting problem that we are currently working on, and feel justifiably lies beyond the scope of the current work. At the very least, the microcanonical approach put forward in ref [18] cannot be naively implemented here since the Fredholm kernel is not integrable when calculating the vertex correlator in a generic state.

---

## Editorial Decision

published